# Effect of Serum SPARC Levels on Survival in Patients with Digestive Tract Cancer: A Post Hoc Analysis of the AMATERASU Randomized Clinical Trial

**DOI:** 10.3390/cancers12061465

**Published:** 2020-06-04

**Authors:** Taisuke Akutsu, Eisaku Ito, Mitsuo Narita, Hironori Ohdaira, Yutaka Suzuki, Mitsuyoshi Urashima

**Affiliations:** 1Division of Molecular Epidemiology, The Jikei University School of Medicine, 3-25-8, Nishi-Shimbashi, Minato-Ku, Tokyo 105-8461, Japan; t-akutsu@jikei.ac.jp; 2Department of Surgery, International University of Health and Welfare Hospital, 537-3 Iguchi, Nasushiobara, Tochigi 329-2763, Japan; i_eisaku_ukasie@yahoo.co.jp (E.I.); ohdaira@iuhw.ac.jp (H.O.); yutaka@iuhw.ac.jp (Y.S.); 3Department of Radiology, International University of Health and Welfare Hospital, 537-3 Iguchi, Nasushiobara, Tochigi 329-2763, Japan; m-narita@iuhw.ac.jp

**Keywords:** SPARC, sarcopenia, muscle, osteonectin, BM-40, esophageal cancer, gastric cancer, colorectal cancer, survival, myokine

## Abstract

Observational studies suggest that physical activity may improve, whereas sarcopenia may worsen the survival of cancer patients. It has been suggested that secreted protein acidic and rich in cysteine (SPARC), one of the myokines that is secreted into the bloodstream by muscle contraction, has tumor-suppressive effects. Based on the hypothesis that serum SPARC level may be a potential prognostic biomarker, a post hoc analysis of the AMATERASU randomized, double-blind, placebo-controlled trial of postoperative oral vitamin D supplementation (2000 IU/day) in patients with stage I–III digestive tract cancer from the esophagus to the rectum (UMIN000001977) was conducted with the aim of exploring the association between serum SPARC levels after operation and survival. On multivariate analyses adjusting serum 25-hydroxyvitamin D, vitamin D supplementation, sarcopenia, body mass index, age, sex, cancer loci, stage, and adjuvant chemotherapy, patients with SPARC levels lower than the median level had a significantly higher risk for death than those with higher levels (hazard ratio, 2.25; 95% confidence interval, 1.25–4.05; *p* = 0.007), whereas there were no significant associations with other outcomes including recurrence. However, on the same multivariate analyses, sarcopenia was not a risk factor for death and/or relapse. These results suggest that serum SPARC levels may be a potential biomarker for death but not cancer relapse.

## 1. Introduction

Observational studies suggest that high physical activity may reduce the risk of sarcopenia, or loss of muscle mass, and may decrease cancer mortality [1,2], but the mechanisms are unknown. As one of the plausible mechanisms, the direct effects of the proteins released from contracting muscle fibers into the bloodstream, so-called myokines, such as secreted protein acidic and rich in cysteine (SPARC), might have a beneficial effect on many organ systems and apoptotic effects on colorectal cancer [3]. SPARC is expressed mainly when tissues undergo changes such as tissue renewal, remodeling, and repair and bind to matrix proteins and control cellular interaction with the extracellular matrix [4]. Aoi et al. suggested that exercise stimulates SPARC secretion from muscle tissues, and that SPARC inhibits colon tumorigenesis by increasing apoptosis [5]. Therefore, we hypothesized that high physical activity suppresses cancer growth by raising serum SPARC levels, and thus higher SPARC levels may be associated with a lower relapse rate and longer relapse-free survival (RFS). In contrast, patients with sarcopenia were hypothesized to have lower SPARC levels, which may thus have an association with shorter RFS. However, few clinical studies have examined the relationship between serum SPARC levels after surgery and the survival of patients with cancer. In addition, it has been suggested that vitamin D stimulates the proliferation and differentiation of skeletal muscle fibers, maintaining and improving muscle strength and physical performance [6].

We and our colleagues previously reported the AMATERSU trial [7], a randomized, clinical trial (RCT) with vitamin D supplementation in digestive tract cancer (esophageal to rectum) patients. Following this research, we also reported our post hoc analysis of the effect modification by cancer pathological subtypes [8], immunohistological characteristics [9], and bioavailable vitamin D levels [10] with vitamin D supplementation on clinical outcomes. The present post hoc analysis was conducted to examine the relationships among serum SPARC levels, psoas muscle index (PMI) values indicating sarcopenia, serum 25-hydroxyvitamin D (25[OH]D) levels, vitamin D supplementation, and relapse or death in these patients.

## 2. Results

### 2.1. Study Population

Of the 417 patients with digestive tract cancers randomly assigned to receive vitamin D supplements (*n* = 251) or placebo (*n* = 166), serum samples were not obtained from 20 patients in the vitamin D group and 17 patients in the placebo group. Thus, 231 (92%) patients in the vitamin D group and 149 (92%) patients in the placebo group, a total of 380 patients, were included in the analysis (Figure 1).

The median follow-up of these 380 patients was 3.5 years (Interquartile range [IQR]: 2.3–5.3 years).

### 2.2. Serum SPARC Level Assessment and Effect of Vitamin D Supplementation

Serum for SPARC level assessment was obtained after surgery and before starting supplementation (i.e., after surgery and at the first outpatient visit). The median serum SPARC level was 478 (IQR, 351–611) ng/mL, and the minimum and maximum values were 92 and 1633 ng/mL, respectively. The serum 25(OH)D level before starting supplementation in the group of patients for whom the SPARC level was measured was 21 ng/mL (IQR: 16–27 ng/mL). The serum SPARC level was not associated with the serum 25(OH)D level (Spearman’s rho = 0.009; *p* = 0.86). A total of 380 patients were divided into two subgroups stratified by the median (478 ng/mL) SPARC level in a post hoc manner.

### 2.3. Skeletal Muscle and Sarcopenia Assessment and Effect of Vitamin D Supplementation

The calculation of PMI was available in 369 (97%) patients. Figure 2 shows the measurement of the psoas muscle used for PMI calculation.

The median PMI value before surgery was 6.2 (IQR: 4.8–7.4) cm^2^/m^2^, and the highest and lowest values were 1.9 and 11.8 cm^2^/m^2^, respectively. There were 124 patients (34%) who met the definition of sarcopenia. The PMI values had a significant moderate association with serum 25(OH)D levels (Spearman’s rho = 0.20; *p* < 0.0001) (Figure 3). 

Excluding patients with sarcopenia at randomization, sarcopenia was diagnosed in 26 (20.2%) patients taking vitamin D and 17 (17.3%) patients taking placebo (RR, 1.16; 95% CI, 0.67–2.02), with no significant difference.

The effects of vitamin D supplementation on the PMI values were evaluated (Figure 4).

Values of PMI were significantly decreased in both the placebo and vitamin D groups 2 years after starting supplementation compared to before surgery. However, the change ratios of PMI were not different between the groups.

### 2.4. Patient Demographics

Baseline patient demographics of the two groups divided by the median serum SPARC levels are shown in Table 1. 

The group with the higher SPARC level was younger, taller, and heavier, and had less esophageal cancer, more colorectal cancer, and more adenocarcinoma than the group with lower SPARC levels. On the other hand, there were no differences between the two groups in the number assigned to vitamin D supplements, the value of serum 25 (OH) D before the intervention, sex, body mass index (BMI), waist circumference, comorbidities, clinical stage, adjuvant chemotherapy, and other chemistry profiles including casual blood glucose levels and hemoglobin A1c.

### 2.5. Relationship Between Serum SPARC Levels and Sarcopenia

Figure 5 shows the serum SPARC levels for either the absence or presence of sarcopenia. The median (IQR) SPARC was 502 (365–618) ng/mL in patients without sarcopenia, which was higher than in patients with sarcopenia at 435 (316–608) ng/mL, although there was no significant difference.

### 2.6. Effects of Serum SPARC Levels and Sarcopenia on Survival

Since vitamin D supplementation had no effects on PMI values and the development of sarcopenia, the following survival analyses were not compared between the vitamin D group and the placebo group, but they were compared either between the SPARC lower and the SPARC higher than median groups, or between the absence and presence of sarcopenia at diagnosis. Nelson–Aalen cumulative hazard curves for death, relapse or death, and relapse by competing risk analysis are shown in Figure 6.

On univariate analysis, hazard ratios (HRs) for death were significant for low SPARC levels and sarcopenia (Table 2). However, on multivariate analysis, HRs for death as the outcome were significant only for low SPARC levels, not sarcopenia. HRs for relapse or death were not significant for low SPARC levels or sarcopenia on univariate and multivariate analyses. In addition, the SHRs for both were not significant for recurrence with death as a competing risk.

## 3. Discussion

Before the analyses, the hypothesis was that physical activity suppresses cancer growth by increasing serum SPARC levels, and thus higher SPARC levels may be associated with a reduced relapse rate and longer RFS. Contrary to this hypothesis, there were no significant associations between relapse/RFS and SPARC levels. However, patients with higher SPARC levels had better overall survival (OS) than those with lower levels, which remained significant even after multivariate adjustment including sarcopenia. Physical activity was found to increase the expression and secretion of SPARC in skeletal muscle in both mice and humans [5]. A large cohort study demonstrated that higher physical activity was associated with a lower risk of all-cause mortality [11]. Another cohort study suggested that one in nine deaths from cancer in inactive individuals could have been averted if they performed 15 min of moderate–intensity daily exercise [12]. In addition, a meta-analysis showed that a higher level of postdiagnosis sedentary behavior was associated with an increased risk of all-cause mortality in patients with cancer [13]. It has been suggested that moderate/vigorous physical activity reduces all-cause mortality by 26%, mortality due to ischemic heart disease by 35%, and mortality due to all cancers by more than 17% [12]. Higher serum SPARC levels were suggested to be a biomarker of muscle contraction [4,5] and shown to be associated with lower OS in this study. However, these may not always suggest direct effect of SPARC molecule on survival, but may have indirect effects through interactions with insulin-like growth factor (IGF), insulin, and blood glucose as well as with inflammatory cytokines. Further research is needed to confirm these mechanisms as future directions.

In contrast, it was hypothesized that patients with sarcopenia have lower SPARC levels, which then may have an association with shorter RFS. There was no significant association between SPARC levels and sarcopenia. Although patients with sarcopenia had significantly worse OS in a univariate model, its significance was lost by adjusting by stage, cancer site, and other variables. These results suggest reverse causality, that sarcopenia is merely a precursor to relapse and death, or that the effect of sarcopenia may be confounded by other prognostic factors such as cancer site and stage. A meta-analysis including 7843 patients from 38 studies demonstrated that sarcopenia at cancer diagnosis was associated with worse survival in patients with solid tumors [2], but further research into understanding the negative effects of sarcopenia in adults with cancer is needed for the above reasons.

Recent studies suggest that vitamin D receptor might be expressed in muscle fibers, and vitamin D signaling via the receptor plays a role in the regulation of myoblast proliferation and differentiation [14]. The effect of oral vitamin D supplementation to prevent or treat sarcopenia was still controversial in RCTs [6]. However, in the present, post hoc analysis of a randomized, double-blind, placebo-controlled trial of vitamin D performed in patients with cancer, vitamin D supplementation had no impact on psoas muscle mass compared with placebo, although the PMI values had a significant and moderate association with serum 25(OH)D levels, suggesting that higher 25(OH)D levels can be largely confounded by healthy lifestyles [15] that include daily physical activity [16,17]. However, the present analysis had a skewed study population, such as postoperative patients with digestive tract cancers, and the sample size required to investigate the effect of maintaining or improving muscle mass was not calculated. For this reason, no conclusions can be drawn as to whether vitamin D supplementation affects muscle mass or sarcopenia.

This analysis has several limitations. First, lifestyle factors such as daily physical activity before and after operation were not included among the clinical variables. Thus, associations between SPARC levels and physical activity could not be determined. Second, we did not measure the molecules involved in glucose tolerance, such as IGF or insulin, and inflammatory cytokines, which are well known to be affected by physical activity [18,19] and operation [20]. Thus, it is difficult to describe how serum SPARC can be related to IGF–insulin axis and affect inflammatory and cytokine markers. Third, major confounders, such as smoking and alcohol consumption, were not measured in this study. Fourth, a sample size calculation for this purpose was not performed, and it was not verified using other independent cohorts. Therefore, it remains possible that the significant differences shown in the present analysis occurred by chance. Fifth, there were cases where no serum sample was available before the intervention. However, the difference from the study population was not large, and it is assumed that the lack of samples occurred randomly. Sixth, there is the possibility of measurement error of the serum SPARC value. All serum samples were stored in the same environment, but since all serum samples were used to measure serum SPARC levels at the end of the study, there may be a difference in quality between older and more recent samples. Seventh, since the AMATERASU trial was conducted in Japan, the patients were Asian, and the results of this analysis are thus not necessarily generalizable to other populations.

## 4. Materials and Methods

### 4.1. Trial Design Overview

This analysis was based on the data of the participants in the AMATERASU trial (UMIN000001977), which is a randomized, double-blind, placebo-controlled trial to investigate the effect of vitamin D supplementation in patients with digestive tract cancers. The AMATERASU trial was conducted at the International University of Health and Welfare Hospital in Japan between January 2010 and February 2018. The ethics committee of the International University of Health and Welfare Hospital (ethics approval code: 13–B–263), as well as the Jikei University School of Medicine (ethics approval code: 21–216(6094)), approved the study protocol. Written, informed consent was obtained from all patients before surgery. The aim of the AMATERASU trial was to determine whether postoperative vitamin D3 supplementation (2000 IU/day) can improve the survival of patients with digestive tract cancers. Details of the study design are described in the original report [7].

### 4.2. Study Population

Eligible patients in the AMATERASU trial were aged 30 to 90 years, had a histopathological diagnosis of stage I to III epithelial carcinoma of the digestive tract (esophagus, stomach, small intestine, colon, or rectum), underwent curative surgery with complete tumor resection, did not take vitamin D supplements or active vitamin D before study enrollment, and had no history of urinary tract stones. Patients were excluded if tumors were not resectable by surgery or serious postoperative complications occurred before starting study drug supplementation. Preoperative and postoperative chemotherapy was administered to patients with stage II and III esophageal cancer. Postoperative chemotherapy was administered to patients with stage II and III gastric cancer and all patients with stage III colorectal cancer. This post hoc analysis included patients who were randomized in the AMATERASU trial for whom post-operative and pre-intervention serum samples were available for assessment.

### 4.3. Outcome Measures

The outcome measures were overall survival (OS), RFS, and relapse. OS was defined as the elapsed time from the date of randomization to the (i.e., time from starting the study medication) date of death due to any cause. RFS was defined as the elapsed time from the date of randomization to the earliest date of cancer relapse or death due to any cause. Relapse was defined as the elapsed time from the date of randomization to the date of cancer relapse; death was treated as a sensor.

### 4.4. Measurement of Serum SPARC Levels

Serum samples were collected after curative surgery and before starting supplementation (i.e., after surgery and before the first outpatient visit). The median duration from operation to sample obtained was 23 days (IQR 13–42 days). The serum samples were stored at −80 °C prior to use. Serum SPARC levels were measured by T.A. using the Human SPARC SimpleStep ELISA Kit (ab220654) (Abcam, Cambridge, MA, USA) according to the manufacturer’s protocol. All serum samples were diluted 1:400 with the supplied solution before being added to the wells. Each sample was tested in duplicate, and the means were used for analysis.

### 4.5. Skeletal Muscle Assessment

The cross-sectional area of the psoas muscle at the caudal end of the L3 level was measured by E.I. and M.N. using SYNAPSE VINCENT software (Fujifilm Medical, Tokyo, Japan) at diagnosis and 2 years after the patients started taking the supplements. L3 psoas muscle cross-sectional area was identified and quantified using Hounsfield unit thresholds (−29 to +150) [21]. The total bilateral psoas area at the L3 level was normalized for height using the following equation:(1)$$PMI\left(\frac{{\mathrm{cm}}^{2}}{m^{2}}\right)=\frac{\mathrm{Area}\;\mathrm{of}\;\mathrm{both}\;\mathrm{psoas}\;\mathrm{muscles}\;\left({\mathrm{cm}}^{2}\right)}{{\left(\mathrm{height}\right)}^{2}\;\left(m^{2}\right)}.$$

The psoas muscle index (PMI) cut-off values for sarcopenia in the present analysis were 6.36 (cm^2^/m^2^) for males and 3.92 (cm^2^/m^2^) for females based on a previous report defining sarcopenia in Asian adults [22]. The definition of sarcopenia was based only on skeletal muscle mass, not muscle weakness and not reduced physical activity, in the present analysis.

### 4.6. Statistical Analysis

Parametric and non-parametric continuous variables were compared by the *t*-test and Mann–Whitney test, respectively. Dichotomous variables were compared between the groups by the chi-squared test. Spearman’s rank correlation, represented as rho, while linear regression was used to quantify the strengths of associations between two continuous variables: rho ≥0.4, strong; 0.4> rho ≥0.2, moderate; and rho <0.2, weak. Paired and unpaired continuous variables were compared by the Wilcoxon signed-rank test and the Mann–Whitney test, respectively. The effects of serum SPARC levels and sarcopenia on the risk of relapse or death, as well as all-cause death, were estimated using Nelson–Aalen cumulative hazard curves for outcomes. A Cox proportional hazards model was used to determine hazard ratios (HRs) and 95% confidence intervals (CIs). The HRs were calculated by univariate analysis and multivariate analysis adjusted for serum SPARC levels, sarcopenia, age, sex, body mass index (BMI), vitamin D supplementation, serum 25(OH)D levels before the supplementation, cancer location, clinical stage, and adjuvant chemotherapy. To evaluate the effects of serum SPARC levels and sarcopenia on relapse, cumulative incidence functions were applied by considering patient deaths due to causes other than relapse as a competing risk, and competing risk regression was performed using the subdistribution hazard ratio (SHR) and 95% CI [23]. Values with two-sided *p* < 0.05 were considered significant. All data were analyzed using Stata 14.0 (StataCorp LP, College Station, TX, USA).

## 5. Conclusions

In conclusion, for postoperative patients with digestive tract cancers, vitamin D supplementation may be less effective for muscle maintenance. Moreover, serum levels of SPARC, known as one of the myokines, may be a predictive biomarker only for death, but not recurrence.

## Figures and Tables

**Figure 1 cancers-12-01465-f001:**
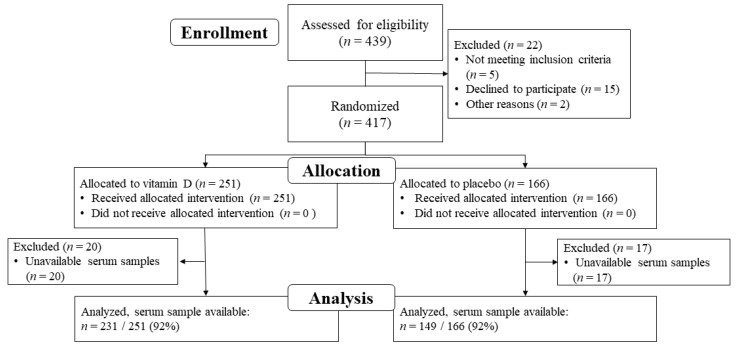
Flow chart of the research participants.

**Figure 2 cancers-12-01465-f002:**
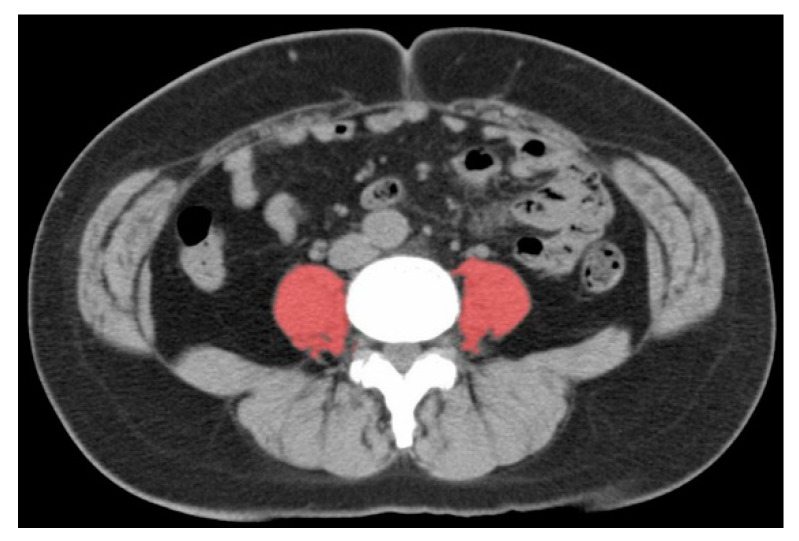
Image of the total bilateral psoas muscle area at the L3 level. The red area is measured as the psoas region.

**Figure 3 cancers-12-01465-f003:**
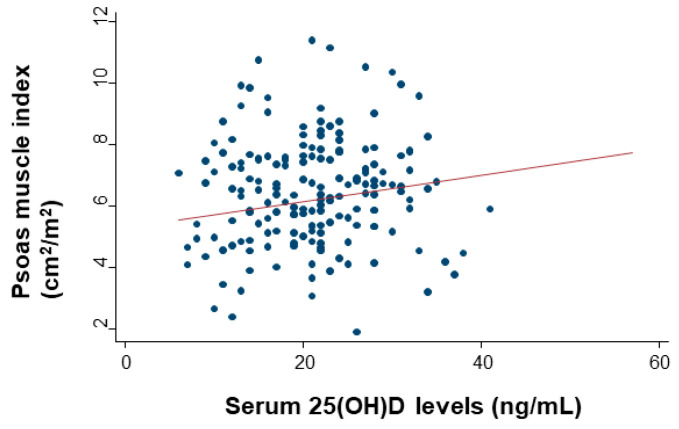
Collinearity levels of serum serum 25-hydroxyvitamin D (25(OH)D) levels and the psoas muscle index (PMI). Spearman’s rank correlation with linear regression was used to quantify the strength of the association.

**Figure 4 cancers-12-01465-f004:**
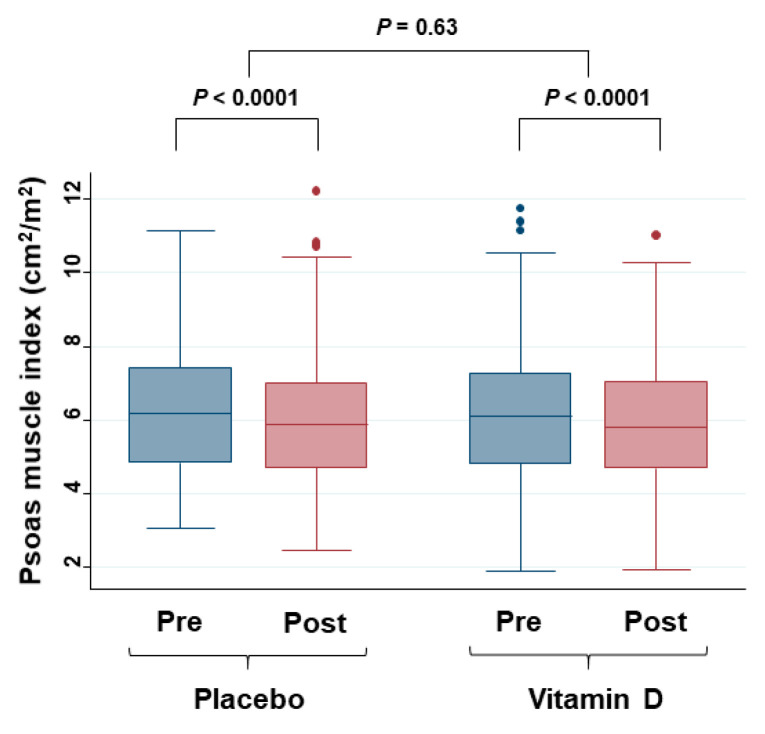
Box plot of changes in serum PMI values in the placebo group and the vitamin D group. Pre = at diagnosis; Post = 2 years after starting supplementation. Changes between pre and post were evaluated with the Wilcoxon signed-rank test, and the change ratio ((post–pre)/pre) was compared between the placebo group and the vitamin D group by the Mann–Whitney test.

**Figure 5 cancers-12-01465-f005:**
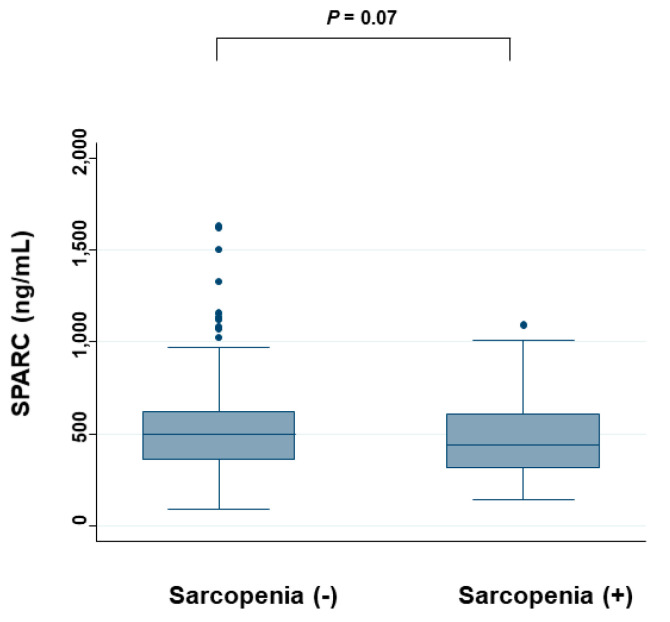
Serum SPARC levels for either the absence or presence of sarcopenia. The SPARC levels are compared between the placebo group and the vitamin D group by the Mann–Whitney test.

**Figure 6 cancers-12-01465-f006:**
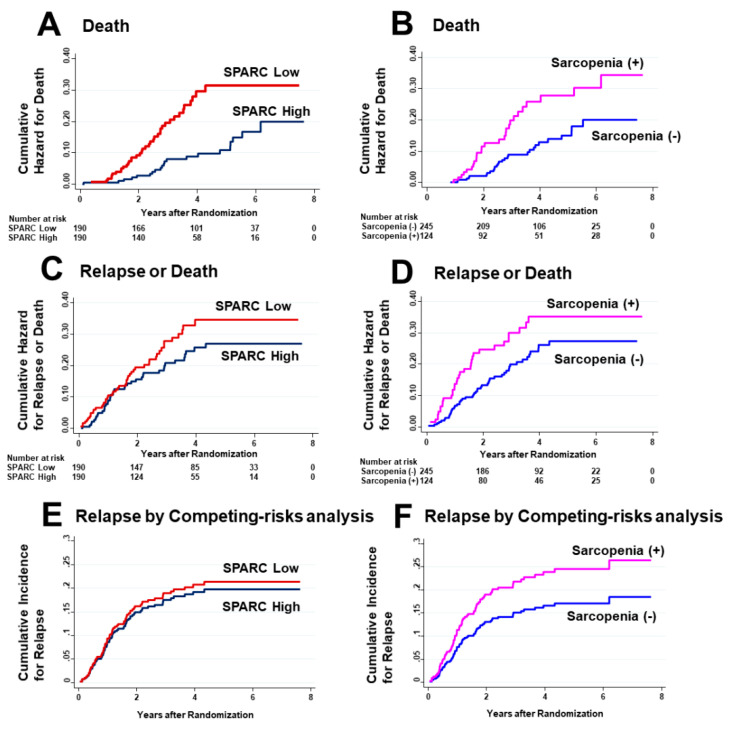
Cumulative hazard curves for outcomes. Cumulative hazard curves for death stratified by SPARC levels (**A**) and absence or presence of sarcopenia (**B**). Cumulative hazard curves for relapse or death stratified by SPARC levels (**C**) and absence or presence of sarcopenia (**D**). Cumulative hazard curves for relapse stratified by SPARC levels (**E**) and absence or presence of sarcopenia (**F**).

**Table 1 cancers-12-01465-t001:** Patient demographic characteristics of all patients and the two groups divided by median serum SPARC levels.

Characteristics	Total (*n* = 380)	SPARC Low (*n* = 190)	SPARC High (*n* = 190)
**Baseline SPARC level**			
ng/mL, median (IQR)	478 (351–611)	351 (275–409)	611 (538–720)
**Intervention**			
Vitamin D supplementation, no. (%)	231 (61)	120 (63)	111 (58)
**Baseline 25(OH)D levels**			
ng/mL, median (IQR)	21 (16–27)	21 (16–28)	21 (16–27)
**Subgroup of 25(OH)D, no. (%)**			
< 20 ng/mL	153 (41)	77 (41)	76 (40)
20–40 ng/mL	219 (58)	108 (58)	111 (59)
> 40 ng/mL	5 (1)	3 (1)	2 (1)
Female, no. (%)	125 (33)	61 (32)	64 (33)
Age (y), mean (SD)	66 ± 11	68 ± 11	64 ± 10
**Physical condition**			
Height (cm), median (IQR)	161 (154–167)	161 (154–167)	162 (155–168)
Weight (kg), median (IQR)	56 (49–63)	55 (48–61)	58 (51–65)
BMI (kg/m^2^), median (IQR)	22 (20–24)	21 (19–23)	22 (20–24)
Waist circumference (cm), median (IQR)	81 (75–86)	80 (74–85)	82 (76–88)
Systolic blood pressure (mmHg), median (IQR)	126 (115–136)	126 (116–135)	126 (115–136)
Diastolic blood pressure (mmHg), median (IQR)	73 (67–81)	72 (66–80)	75 (68–82)
**Comorbid conditions, no. (%)**			
Hypertension	143 (38)	71 (37)	72 (38)
Diabetes mellitus	61 (16)	30 (16)	31 (16)
Endocrine disease	41 (11)	19 (10)	22 (12)
Cardiovascular disease	28 (7)	15 (8)	13 (7)
Chronic kidney disease	4 (1)	4 (2)	0 (0)
Asthma	3 (1)	1 (1)	2 (1)
Orthopedic disease	2 (1)	1 (1)	1 (1)
History of other cancers	15 (4)	12 (6)	3 (2)
**Site of cancer, no. (%)**			
Esophageal cancer	37 (10)	27 (14)	10 (5)
Gastric cancer	167 (44)	84 (44)	83 (44)
Small bowel cancer	2 (1)	1 (1)	1 (1)
Colorectal cancer	174 (46)	78 (41)	96 (51)
**Clinical stage, no. (%)**			
Stage I	169 (44)	81 (43)	88 (46)
Stage II	100 (26)	54 (28)	46 (24)
Stage III	111 (29)	55 (29)	56 (29)
**Pathological subtype, no. (%)**			
Adenocarcinoma	340 (89)	160 (84)	180 (95)
Squamous cell carcinoma	35 (9)	26 (14)	9 (5)
Others	5 (1)	4 (2)	1 (1)
**Adjuvant chemotherapy, no. (%)**	133 (35)	70 (36)	63 (33)
**Baseline serum levels, median (IQR)**			
Calcium (mg/dL)	9.3 (9.0–9.6)	9.2 (8.9–9.5)	9.4 (9.0–9.6)
ALP (IU/L)	228 (187–265)	222 (187–262)	234 (187–270)
Intact PTH (pg/mL)	40 (32–53)	43 (33–55)	38 (31–50)
Total cholesterol (mg/dL)	186 (161–209)	181 (160–202)	191 (168–212)
High-density cholesterol (mg/dL)	50 (41–60)	51 (40–59)	49 (40–60)
Triglycerides (mg/dL)	101 (73–140)	97 (70–136)	106 (77–145)
Blood glucose (mg/dL)	103 (94–117)	101 (92–121)	104 (96–115)
HbA1c (%)	5.6 (5.2–5.9)	5.6 (5.2–5.9)	5.6 (5.3–5.9)
BUN (mg/dL)	14 (11–16)	14 (11–17)	13 (11–16)
Creatinine (mg/dL)	0.8 (0.6–0.9)	0.8 (0.6–0.9)	0.8 (0.6–0.9)

Percentages may not equal 100% due to rounding. SPARC, secreted protein acidic and rich in cysteine; IQR, interquartile range; 25(OH)D, 25-hydroxyvitamin D; SD, standard deviation; BMI, body mass index (weight [kg]/height squared [m^2^]); ALP, alkaline phosphatase; Intact PTH, the biologically active form of parathyroid hormone; BUN, blood urea nitrogen.

**Table 2 cancers-12-01465-t002:** Univariate and multivariate analyses of serum SPARC levels or sarcopenia for outcomes.

Analysis Type	Death	*p* Value	Relapse or Death	*p* Value	Relapse	*p* Value
**SPARC low vs. high, HR (95% CI)**						
Univariate analysis	2.37 (1.38–4.04)	0.002	1.30 (0.85–1.98)	0.22	1.10 (0.69–1.74)	0.70
Multivariate analysis	2.25 (1.25–4.05)	0.007	1.20 (0.76–1.89)	0.43	0.92 (0.57–1.51)	0.75
**Sarcopenia presence vs. absence, HR (95% CI)**						
Univariate analysis	1.77 (1.05–2.99)	0.03	1.42 (0.93–2.17)	0.11	1.51 (0.95–2.40)	0.08
Multivariate analysis	1.30 (0.72–2.35)	0.38	1.06 (0.66–1.70)	0.82	1.18 (0.69–2.00)	0.55

The subdistribution hazard ratio (SHR) was calculated only for competing risk regression. The hazard ratio (HR) was adjusted for serum SPARC levels, sarcopenia, age, sex, BMI, vitamin D supplement intervention, serum 25[OH]D levels before the intervention, cancer location, clinical stage, and adjuvant chemotherapy. SPARC, secreted protein acidic and rich in cysteine; CI, confidence interval; 25(OH)D, 25-hydroxyvitamin D; BMI, body mass index (weight [kg]/height squared [m^2^]).

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
