# Peer review of "Effect of Serum SPARC Levels on Survival in Patients with Digestive Tract Cancer: A Post Hoc Analysis of the AMATERASU Randomized Clinical Trial"

_cancers, 2020, doi:10.3390/cancers12061465_

Round 1

Reviewer 1 Report

The article can be published in the present form, in my opinion.

Reviewer 2 Report

This is a post hoc analysis of the effect of serum SPARC levels on survival in patients with digestive tract cancer within a randomized controlled trial.

This is a well written paper with appropriate study design within the confines of a post hoc analyses. Ideally, the study will be more robust if the multivariate models can be adjusted for other confounders such as smoking, alcohol, physical activity, components of IGF axis and inflammatory markers. I do however note that the authors have acknowledged the various limitations of their study.

Minor comments

  1. Suggest changing "Under the hypothesis that the serum SPARC level is a novel biomarker" to "Based on the hypothesis that serum SPARC level may be a potential prognostic biomarker"
  2. "These results suggest that serum SPARC levels may be a biomarker for death but not relapse" to "These results suggest that serum SPARC levels may be a potential biomarker for death but not cancer relapse".

Author Response

This manuscript is a resubmission of an earlier submission. The following is a list of the peer review reports and author responses from that submission.

Round 1

Reviewer 1 Report

The article “Effect of serum SPARC levels on survival in patients 2 with digestive tract cancer: A post hoc study of the 3 AMATERASU randomized clinical trial” by Akutsu and colab. is interesting and suggests an additional biomarker for assessing cancer patient’s risk. However, the writing style is somewhat confusing and I found it difficult to read. After re-reading the original AMATERASU trial the article became easier to assess, but I believe that readers should not have to read another article just to make the article they are reading easier to understand.

  1. The keywords could be improved – SPARC should definitely be included in the keywords and perhaps prognosis could be excluded
  2. In the Results section, rows 70-74, you report median serum SPARC levels, but you do not explain in which group or if it was in samples taken before or after surgery. What was the serum level of vitamin D in these patients? When did you divide the 380 patients according to SPARC level? (during the trial or in the post-hoc analysis)
  3. How long was the follow up for these patients? Did you assess SPARC levels at the follow-up visits?
  4. Page 3, rows 90-92 “The effects of vitamin D supplementation on the PMI values were evaluated (Figure 4). Values of PMI were significantly decreased in both the placebo and vitamin D groups compared between pre and post starting supplementation. However, the change ratios of PMI were not different between the groups.” How long after randomization did this happen?
  5. In the Material and method section, you state that inclusion criteria were “did not take vitamin D supplements or active vitamin D”. However, in the statistical analysis section, you say you adjusted for vitamin D intervention...please reconcile and clearly define your study population.
  6. In the Material and Method section, you state that the outcome measures were “overall survival (OS), RFS, and relapse”. You then proceed to defining the first two, but you do not define what you mean by “relapse” and how you quantify it and in what way it is different from RFS

Minor comments:

  1. Abstract – rows 27-29 “ (...) had a significantly higher risk for death than those with higher levels (hazard ratio, 2.25; 95% confidence interval, 1.25-4.05; P=0.007), whereas there were no significant associations with relapse or death and relapse alone.” the statement is confusing. Please rephrase in order to increase clarity
  2. Page 8, row 202 – you refer to your work as a “study” when in fact it is a retrospective analysis. Please replace “study” with analysis in this instance and wherever else you describe your work
  3. Page 8, row 225 – “Serum samples were collected before and after curative surgery” instead of ”Serum samples were collected after curative surgery and before the intervention”. Also, you should mention exactly how long before and after surgery you collected the samples

Reviewer 2 Report

This is a post hoc study evaluating the association of post operative serum SPARC and survival in patients with digestive tract cancer. Overall, the paper is well written. 

My comments and suggestions as follows:

  1. Multivariate analyses should include confounders such as smoking, alcohol intake, adjuvant treatment (any post op radiotherapy or chemoradiation in addition to adjuvant chemotherapy)
  2. Can the authors explain or hypothesize why serum SPARC is positively associated with death but not relapse or death/relapse?
  3. I would recommend that the authors investigate the correlation between serum SPARC and SPARC measurement in tumor tissues? There have been multiple studies reporting association between SPARC levels in tissue and cancer prognosis.
  4. Since the authors' hypothesis is that physical activity leads to changes in SPARC levels which affect cancer prognosis, I would recommend that the authors measure other biomarkers e.g. insulin, glucose, IGF, adipokines, inflammatory markers (CRP, interleukins, tumor necrosis factor) which are all tightly linked to physical activity and sarcopenia
  5. Can the authors elaborate on the duration from operation that the post operative samples were obtained?

Round 2

Reviewer 2 Report

I thank the authors for performing additional analyses and for the comments. I note the inability to account for confounders such as smoking, alcohol and radiotherapy due to a lack of information.

The authors cited the inability to measure other biomarkers due to a lack of timing of food intake. Although there is a lack of timing of food intake, I think it will be useful to measure these biomarkers. It is known that physical activity and operation affects the IGF-insulin axis and affect inflammatory and cytokine markers. Hence, these biomarkers may affect prognosis and affect the overall results.  

Round 3

Reviewer 2 Report

I note that the authors have acknowledge several limitations in their study including confounders and inability to account for other biomarkers including inflammatory cytokines.